# Replication Study: The CD47-signal regulatory protein alpha (SIRPa) interaction is a therapeutic target for human solid tumors

Stephen K Horrigan, Reproducibility Project: Cancer Biology*

Noble Life Sciences, Gaithersburg, United States

**Abstract** In 2015, as part of the Reproducibility Project: Cancer Biology, we published a Registered Report (Chroscinski et al., 2015) that described how we intended to replicate selected experiments from the paper "The CD47-signal regulatory protein alpha (SIRPa) interaction is a therapeutic target for human solid tumors "(Willingham et al., 2012). Here we report the results of those experiments. We found that treatment of immune competent mice bearing orthotopic breast tumors with anti-mouse CD47 antibodies resulted in short-term anemia compared to controls, consistent with the previously described function of CD47 in normal phagocytosis of aging red blood cells and results reported in the original study (Table S4; Willingham et al., 2012). The weight of tumors after 30 days administration of anti-CD47 antibodies or IgG isotype control were not found to be statistically different, whereas the original study reported inhibition of tumor growth with anti-CD47 treatment (Figure 6A,B; Willingham et al., 2012). However, our efforts to replicate this experiment were confounded because spontaneous regression of tumors occurred in several of the mice. Additionally, the excised tumors were scored for inflammatory cell infiltrates. We found IgG and anti-CD47 treated tumors resulted in minimal to moderate lymphocytic infiltrate, while the original study observed sparse lymphocytic infiltrate in IgG-treated tumors and increased inflammatory cell infiltrates in anti-CD47 treated tumors (Figure 6C; Willingham et al., 2012). Furthermore, we observed neutrophilic infiltration was slightly increased in anti-CD47 treated tumors compared to IgG control. Finally, we report a meta-analysis of the result.

*For correspondence: tim@cos.io; nicole@scienceexchange.com

Group author details: Reproducibility Project: Cancer Biology See page 10

## Introduction

The Reproducibility Project: Cancer Biology (RP:CB) is a collaboration between the Center for Open Science and Science Exchange that seeks to address concerns about reproducibility in scientific research by conducting replications of selected experiments from a number of high-profile papers in the field of cancer biology (*Errington et al., 2014*). For each of these papers a Registered Report detailing the proposed experimental designs and protocols for the replications was peer reviewed and published prior to data collection. The present paper is a Replication Study that reports the results of the replication experiments detailed in the Registered Report (*Chroscinski et al., 2015*) for a paper by Willingham et al., and uses a number of approaches to compare the outcomes of the original experiments and the replications.

In 2012, Willingham et al. reported that blocking the signal regulatory protein alpha (SIRPa)/ CD47 interaction with an anti-CD47 blocking antibody promoted phagocytosis of solid tumor cells *in vitro* and reduced growth of solid tumors *in vivo* indicating that anti-CD47 antibody therapy may be an effective treatment for a variety of solid tumors. Using a syngeneic breast cancer model, mouse anti-CD47 antibody treatment resulted in a statistically significant decrease in final tumor weight compared to IgG isotype control (*Willingham et al., 2012*). Anti-CD47 treatment also increased

lymphocytic infiltration to the tumor site without unacceptable toxicity except short-term anemia observed immediately after dosing.

The Registered Report for the paper by Willingham et al. described the experiments to be replicated (Figure 6A–C and Table S4), and summarized the current evidence for these findings (*Chroscinski et al., 2015*). Since that publication there have been additional studies examining the safety and efficacy of targeting CD47 as an anti-cancer therapeutic. Anti-CD47 treatment was reported to increase macrophage phagocytosis, decrease tumor weight, and inhibit spontaneous metastasis in a osteosarcoma xenograft model (*Xu et al., 2015*). Similarly, CD47 blockade was reported to enhance tumor cell phagocytosis by macrophages, reduce tumor burden, and increase survival in glioblastoma (*Zhang et al., 2016*), gastric cancer (*Yoshida et al., 2015*), and pancreatic neuroendocrine tumor (*Krampitz et al., 2016*) xenograft models. Cioffi and colleagues tested the effect of inhibiting CD47 in pancreatic ductal adenocarcinoma (PDAC) and reported that while anti-CD47 antibodies increased phagocytosis *in vitro*, it did not result in a statistically significant change in tumor growth in a PDAC patient-derived xenograft (PDX) model unless administered in combination with a chemotherapeutic agent (*Cioffi et al., 2015*). Additionally, a humanized anti-CD47 antibody was tested for safety and efficacy in disease models of acute myeloid leukemia (AML) and was reported to decrease tumor burden and increase survival in an AML PDX model (*Liu et al., 2015*). A pre-clinical toxicokinetic study in non-human primates reported no adverse effects associated with the humanized antibody (*Liu et al., 2015*) and patients with AML and solid tumors are being recruited for phase one clinical trials (ClinicalTrials.gov identifiers: NCT02678338 and NCT02216409).

The outcome measures reported in this Replication Study will be aggregated with those from the other Replication Studies to create a dataset that will be examined to provide evidence about reproducibility of cancer biology research, and to identify factors that influence reproducibility more generally.

## Results and discussion

### Engraftment of mouse breast cancer cells and treatment with CD47 targeting antibodies

We sought to independently replicate the safety and efficacy of targeting CD47 in immune competent mice using a syngeneic model of breast cancer. This experiment is similar to what was reported in Figure 6A–C of *Willingham et al. (2012)*. MT1A2 mouse breast cancer cells (*Addison et al., 1995*) were engrafted into the mammary fat pad of syngeneic FVB mice and monitored until palpable tumors formed. Mice with palpable tumors were randomized to receive injections of either 400 μg mouse IgG isotype control (IgG) or 400 μg anti-mouse CD47 (anti-CD47) antibodies every other day into the mammary fat pad proximal to the tumor. While the original study included two clones of anti-CD47 (MIAP410 [*Han et al., 2000*] and MIAP301 [*Lindberg et al., 1993*]), this replication attempt was restricted to only one clone, MIAP410, which had the larger reported effect size of the two clones in the original experiment (*Willingham et al., 2012*).

To test any potential toxicity of the antibody treatment, hematological analysis was performed on blood collected by retro-orbital bleeding 5 days after the beginning of antibody injections. Untreated FVB mice were used to determine the baseline reading. This differed from the original study, which analyzed toxicity in different mouse strains and treatment regimens. In the original study, BALB/c mice were analyzed for blood toxicity 5 days after two 500 μg antibody injections (Table S4) and C57BL/6 mice were analyzed for specific hematology parameters 1, 3, and 6 days after a single intraperitoneal injection of 250 μg of IgG or anti-CD47 antibodies (Supplemental Figure 6) (*Willingham et al., 2012*). Similar to what was reported for C57BL/6 mice in the original paper, FVB mice treated with anti-CD47 resulted in short-term anemia (*Figure 1*; *Figure 1—figure supplement 1*). Red blood cell count (*Figure 1G*; one-way ANOVA; $F_{(2,17)} = 424.9$, uncorrected $p=3.07\times10^{-15}$ with alpha level = 0.0033; (Bonferroni corrected $p=4.60\times10^{-14}$)), hemoglobin (*Figure 1H*; one-way ANOVA; $F_{(2,17)} = 502.1$, uncorrected $p=7.61\times10^{-16}$ with alpha level = 0.0033; (Bonferroni corrected $p=1.14\times10^{-14}$)), and hematocrit (*Figure 1I*; one-way ANOVA; $F_{(2,17)} = 283.0$, uncorrected $p=8.93\times10^{-14}$ with alpha level = 0.0033; (Bonferroni corrected $p=1.34\times10^{-12}$)) were all slightly reduced in anti-CD47 treated mice compared to untreated and IgG treated mice. This is

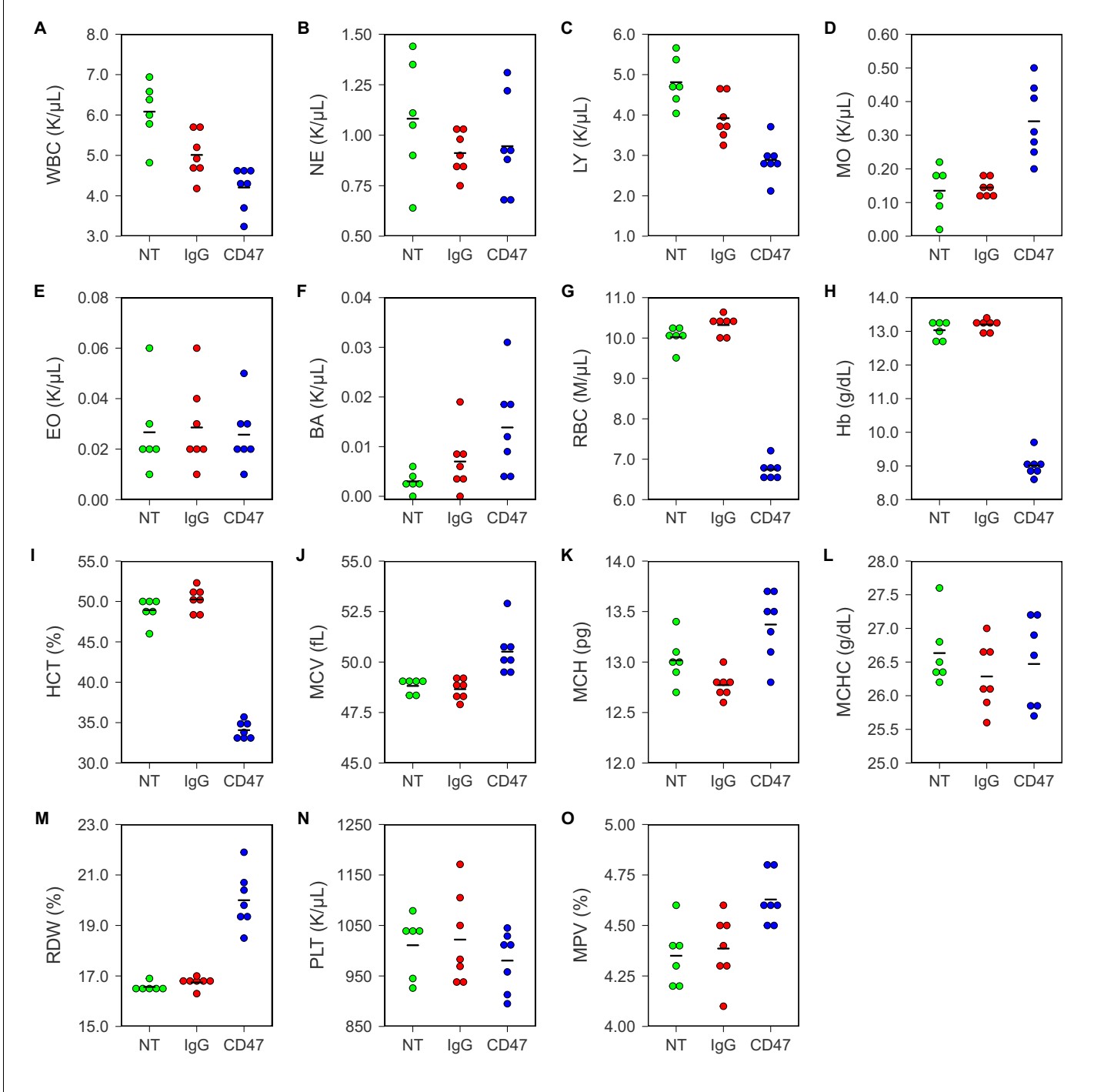

**Figure 1.** Blood toxicity analysis. Female FVB mice bearing orthotopic MT1A2 breast tumors were randomized to receive IgG isotype control (IgG) (n = 7) or anti-mouse CD47 (CD47) (n = 7) antibodies. Mice bearing small or undetectable tumors were designated for baseline reading (NT) (n = 6). Five days after the beginning of treatment blood samples collected via retro-orbital bleed were analyzed on a hematology analyzer. Dot plots with means reported as crossbars for each hematological parameter. For each parameter a one-way ANOVA was performed and the alpha level or *p*-value was adjusted using the Bonferroni correction. (A) White blood cells (WBC), one-way ANOVA; $F(2,17) = 15.09$, uncorrected $p=0.00017$ with alpha level = 0.0033; (Bonferroni corrected $p=0.0026$). (B) Neutrophils (NE), one-way ANOVA; $F(2,17) = 1.02$, uncorrected $p=0.381$ with alpha level = 0.0033; (Bonferroni corrected $p>0.99$). (C) Lymphocytes (LY), one-way ANOVA; $F(2,17) = 20.84$, uncorrected $p=2.67\times10^{-5}$ with alpha level = 0.0033; (Bonferroni corrected $p=0.00040$). (D) Monocytes (MO), Welch's one-way ANOVA; $F(2,8.52) = 9.98$, uncorrected $p=0.0058$ with alpha level = 0.0033; (Bonferroni corrected $p=0.0877$). (E) Eosinophils (EO), one-way ANOVA; $F(2,17) = 0.06$, uncorrected $p=0.942$ with alpha level = 0.0033; (Bonferroni corrected $p>0.99$). (F) Basophils (BA), one-way ANOVA; $F(2,17) = 4.20$, uncorrected $p=0.0330$ with alpha level = 0.0033; (Bonferroni corrected $p=0.495$). (G) Red

*Figure 1 continued on next page*

*Figure 1 continued*

blood cells (RBC), one-way ANOVA; $F_{(2,17)} = 424.9$, uncorrected $p=3.07 \times 10^{-15}$ with alpha level = 0.0033; (Bonferroni corrected $p=4.60 \times 10^{-14}$). (H) Hemoglobin (Hb), one-way ANOVA; $F_{(2,17)} = 502.1$, uncorrected $p=7.61 \times 10^{-16}$ with alpha level = 0.0033; (Bonferroni corrected $p=1.14 \times 10^{-14}$). (I) Hematocrit (HCT), one-way ANOVA; $F_{(2,17)} = 283.0$, uncorrected $p=8.93 \times 10^{-14}$ with alpha level = 0.0033; (Bonferroni corrected $p=1.34 \times 10^{-12}$). (J) Mean corpuscular volume (MCV), one-way ANOVA; $F_{(2,17)} = 11.81$, uncorrected $p=0.00061$ with alpha level = 0.0033; (Bonferroni corrected $p=0.0091$). (K) Mean corpuscular hemoglobin (MCH), one-way ANOVA; $F_{(2,17)} = 10.64$, uncorrected $p=0.00101$ with alpha level = 0.0033; (Bonferroni corrected $p=0.0151$). (L) Mean corpuscular hemoglobin concentration (MCHC), one-way ANOVA; $F_{(2,17)} = 0.61$, uncorrected $p=0.552$ with alpha level = 0.0033; (Bonferroni corrected $p>0.99$). (M) Red blood cell distribution width (RDW), Welch's one-way ANOVA; $F_{(2,10.46)} = 30.62$, uncorrected $p=4.25 \times 10^{-5}$ with alpha level = 0.0033; (Bonferroni corrected $p=0.00064$). (N) Platelets (PLT), one-way ANOVA; $F_{(2,17)} = 0.62$, uncorrected $p=0.548$ with alpha level = 0.0033; (Bonferroni corrected $p>0.99$). (O) Mean platelet volume (MPV), one-way ANOVA; $F_{(2,17)} = 6.98$, uncorrected $p=0.0061$ with alpha level = 0.0033; (Bonferroni corrected $p=0.092$). Additional details for this experiment can be found at https://osf.io/g57ch/.

The following figure supplement is available for figure 1:

**Figure supplement 1.** Blood toxicity analysis.

consistent with the previously described function of CD47 in the normal phagocytosis of aging red blood cells (*Oldenborg et al., 2000*; *Oldenborg et al., 2001*; *Oldenborg, 2004*) and has been observed in other studies examining anti-CD47 antibody treatment (*Liu et al., 2015*). Additionally, three animals treated with anti-CD47 showed mild monocytosis (*Figure 1D*; *Figure 1—figure supplement 1*).

After 30 days of antibody treatment, tumors were excised and weighed (*Figure 2*). Tumors treated with IgG grew to an average of 0.075 grams [n = 7, *SD* = 0.078], while tumors treated with anti-CD47 resulted in an average weight of 0.163 grams [n = 6, *SD* = 0.096]. The comparison of these two groups was not statistically significant (Welch's *t*-test; $t_{(9.66)} = 1.796$, $p=0.104$). This is in comparison to the original study, which reported an average weight of 0.144 grams [n = 5, *SD* = 0.052] for IgG treated tumors and an average of 0.012 grams [n = 5, *SD* = 0.002] in anti-CD47 treated tumors. The range of observed tumor weights in the original study varied from 9 to 198 mg, with IgG treated tumors representing the higher observed weights (60–198 mg) while anti-CD47 treated tumors were reported between 9 and 14 mg. This compares to this replication attempt which observed tumor weights ranging from 5 to 257 mg, with IgG treated tumors (5–203 mg) and anti-CD47 treated tumors (41–257 mg) having fairly similar wide distributions. Indeed, the relative standard deviation (RSD) associated with this replication attempt (IgG treated =104%; anti-CD47 treated =59%) was larger than the RSD reported in the original study (IgG treated =36%; anti-CD47 treated =18%). The RSD of the IgG treated tumors reported in *Willingham et al. (2012)* is similar to the estimated RSDs (~30%) in the control conditions from two other published studies that utilized MT1A2 cells (*Ahn and Brown, 2008*; *Noblitt et al., 2005*), granted these studies injected more cells and in different sites than the original study and this replication attempt. Interestingly, a more recent paper briefly stated that they purposefully did not utilize the MT1A2 cell line in their study because of a high prevalence of spontaneous tumor regression, confounding the results (*Desilva et al., 2012*). An evaluation of tumor growth revealed this also occurred in this replication attempt with three tumors regressing at the end of the study compared to the last tumor volume measurement taken 14 days after the start of treatment (*Figure 2—figure supplement 1*). While these observations confound the results of this replication attempt, we further explored the tumor weight data by conducting the same analysis above, but with the three tumors that regressed during the course of the study removed. This was also not statistically significant (Welch's *t*-test; $t_{(7.94)} = 0.745$, $p=0.478$, Glass'$\Delta = -0.58$, 95% CI [−1.88, 0.80]).

Dissected tumors were further processed and H&E-stained histological sections were blindly analyzed for the extent of lymphocytic infiltration. This is similar to the original study, however this replication attempt used a predefined scoring system to assess the degree of lymphocytic infiltrate (*Demaria et al., 2001*). Both IgG and anti-CD47 treated tumors resulted in minimal to moderate lymphocytic infiltrate (*Table 1*). Although not planned, the tumors were also analyzed for the extent of neutrophilic infiltration. The neutrophilic infiltrate in IgG treated tumors was minimal in 4 and moderate in 3 tumors. The neutrophilic infiltrate in anti-CD47 treated tumors were minimal in 1, moderate in 3, and brisk in 2 tumors (*Table 1*).

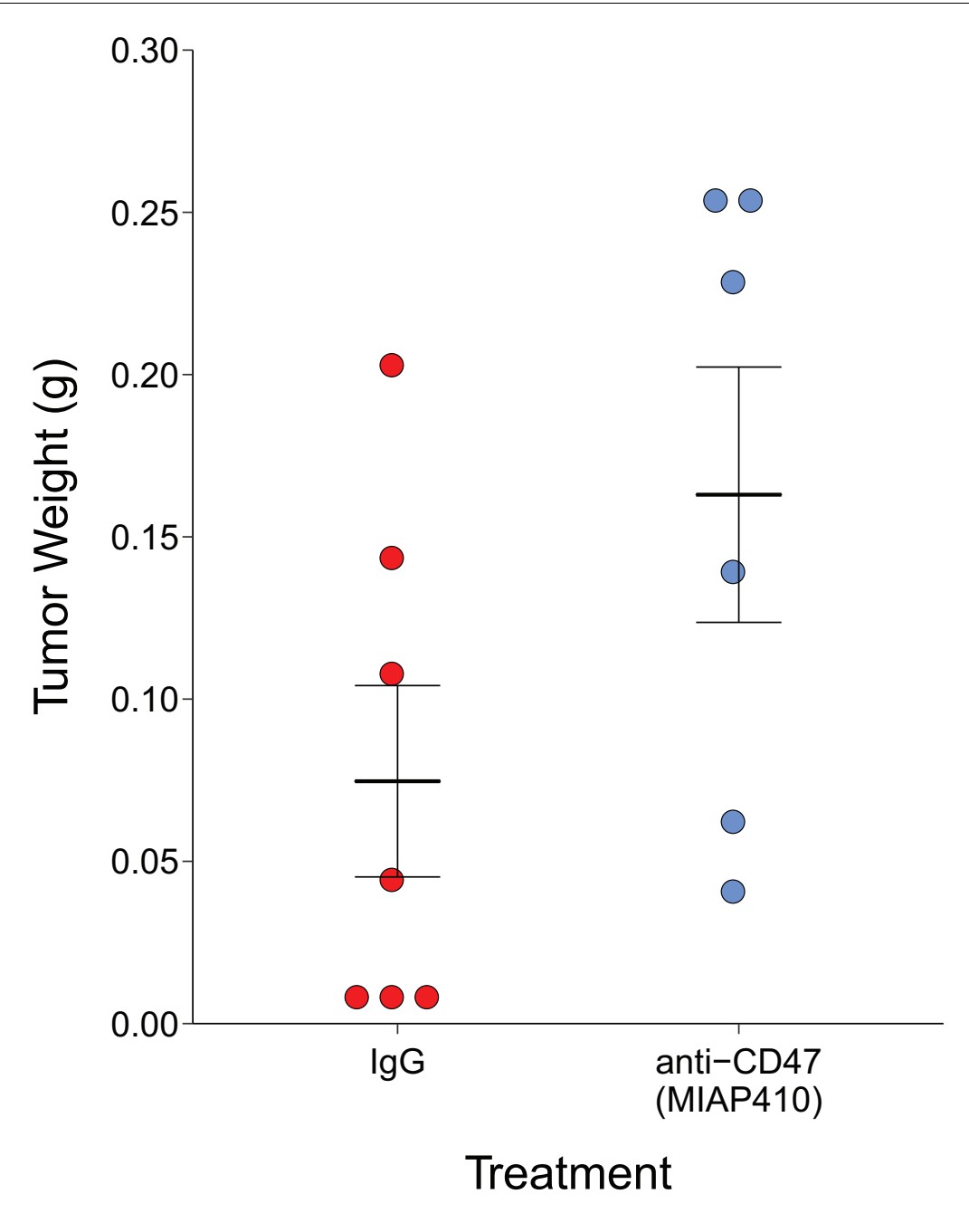

**Figure 2.** Final tumor weights of immune competent hosts treated with control or CD47 targeted antibodies. At the end of the predefined study period (Day 31), tumors from mice bearing orthotopic MT1A2 breast tumors treated every other day with IgG isotype control (IgG) (n = 7) or anti-mouse CD47 (anti-CD47) (n = 6) antibodies were excised and weighed. Dot plot with means reported as crossbars and error bars represent s.e.m. Two-tailed Welch's t-test between IgG and anti-CD47 treated tumors; $t(9.66) = 1.796$, $p=0.104$. Additional details for this experiment can be found at https://osf.io/g57ch/.

The following figure supplement is available for figure 2:

**Figure supplement 1.** Tumor volumes of immune competent hosts treated with control or CD47 targeted antibodies.

**Table 1.** Severity of inflammatory cell infiltration of tumors. Excised tumors were fixed, sectioned, and stained with hematoxylin and eosin and blindly scored by a Board Certified pathologist utilizing the severity score for inflammatory cell infiltrates (**Demaria et al., 2001**). Tumor infiltrating lymphocytes and neutrophils were scored for tumors from mice bearing orthotopic MT1A2 breast tumors treated every other day with IgG isotype control (IgG) (n = 7) or anti-mouse CD47 (CD47) (n = 6) antibodies. Additional details for this experiment can be found at https://osf.io/g57ch/.

| Treatment | Lymphocytic infiltrate | | | | Neutrophilic infiltrate | | | |
| | Absent | Minimal | Moderate | Brisk | Absent | Minimal | Moderate | Brisk |
|---|---|---|---|---|---|---|---|---|
| IgG | 0 | 6 | 1 | 0 | 0 | 4 | 3 | 0 |
| CD47 | 0 | 5 | 1 | 0 | 0 | 1 | 3 | 2 |

There are a number of factors that can affect tumor growth. While tumor growth is exponential under an ideal scenario, factors such as availability of nutrients, oxygen, and space influence and alter the growth of the tumor initially compared to the continued growth of the tumor (**Cornelis et al., 2013**; **Talkington and Durrett, 2015**). Simultaneously, other murine immunogenic tumor models are known to spontaneously regress (**Penichet et al., 2001**; **Robinson et al., 2009**; **Vince et al., 2004**), which is a phenomenon known to naturally occur in cancer patients (**Jessy, 2011**; **Saleh et al., 2005**; **Salman, 2016**) .

## Meta-analysis of original and replicated effects

We performed a meta-analysis using a random-effects model for the effect described above as pre-specified in the confirmatory analysis plan (**Chroscinski et al., 2015**). To provide a standardized measure of the effect, a common effect size Glass' Δ was calculated for the original and replication study. Glass' Δ is the standardized difference between two means using the standard deviation of only the control group. It is used in this case because of the unequal variance between the control and treatment conditions in the original study.

The comparison of IgG treated tumors compared to anti-CD47 treated tumors resulted in Glass' Δ = 2.54, 95% CI [0.40, 4.60] for the data reported in Figure 6B of the original study (**Willingham et al., 2012**). This compares to Glass'Δ = −1.13, 95% CI [−2.35, 0.16] reported in this study. A meta-analysis (**Figure 3**) of these two effects resulted in Glass'Δ = 0.60, 95% CI [−3.00, 4.19], $p$=0.745. The effects for each study are in opposite directions and the point estimate of the replication effect size is not within the confidence interval of the original result, or vice versa. The random effects meta-analysis did not result in a statistically significant effect. Further, the Cochran's $Q$ test for heterogeneity was statistically significant ($p$=0.0039), which along with a large confidence interval around the weighted average effect size from the meta-analysis suggests heterogeneity between the original and replication studies.

This direct replication provides an opportunity to understand the present evidence of these effects. Any known differences, including reagents and protocol differences, were identified prior to conducting the experimental work and described in the Registered Report (**Chroscinski et al., 2015**). However, this is limited to what was obtainable from the original paper and through communication with the original authors, which means there might be particular features of the original experimental protocol that could be critical, but unidentified. So while some aspects, such as cell line, strain and sex of mice, number of cells injected, and the injection site of antibody treatment were maintained, others were unknown or not easily controlled for. These include variables such as cell line genetic drift (**Hughes et al., 2007**; **Kleensang et al., 2016**), circadian biological responses to therapy (**Fu and Kettner, 2013**), mouse strain stocks (**Clayton and Collins, 2014**), housing temperature in mouse facilities (**Kokolus et al., 2013**), and the obesity and microbiome of recipient mice (**Klevorn and Teague, 2016**; **Macpherson and McCoy, 2015**). Additionally, a differential response to immunotherapy can occur due to heterogeneity in individual tumor microenvironments (**Grosso and Jure-Kunkel, 2013**), which has also been observed in the clinical setting (**Ascierto and Marincola, 2014**; **Stevenson, 2014**). Whether these or other factors influence the outcomes of this study is open to hypothesizing and further investigation, which is facilitated by direct replications and transparent reporting.

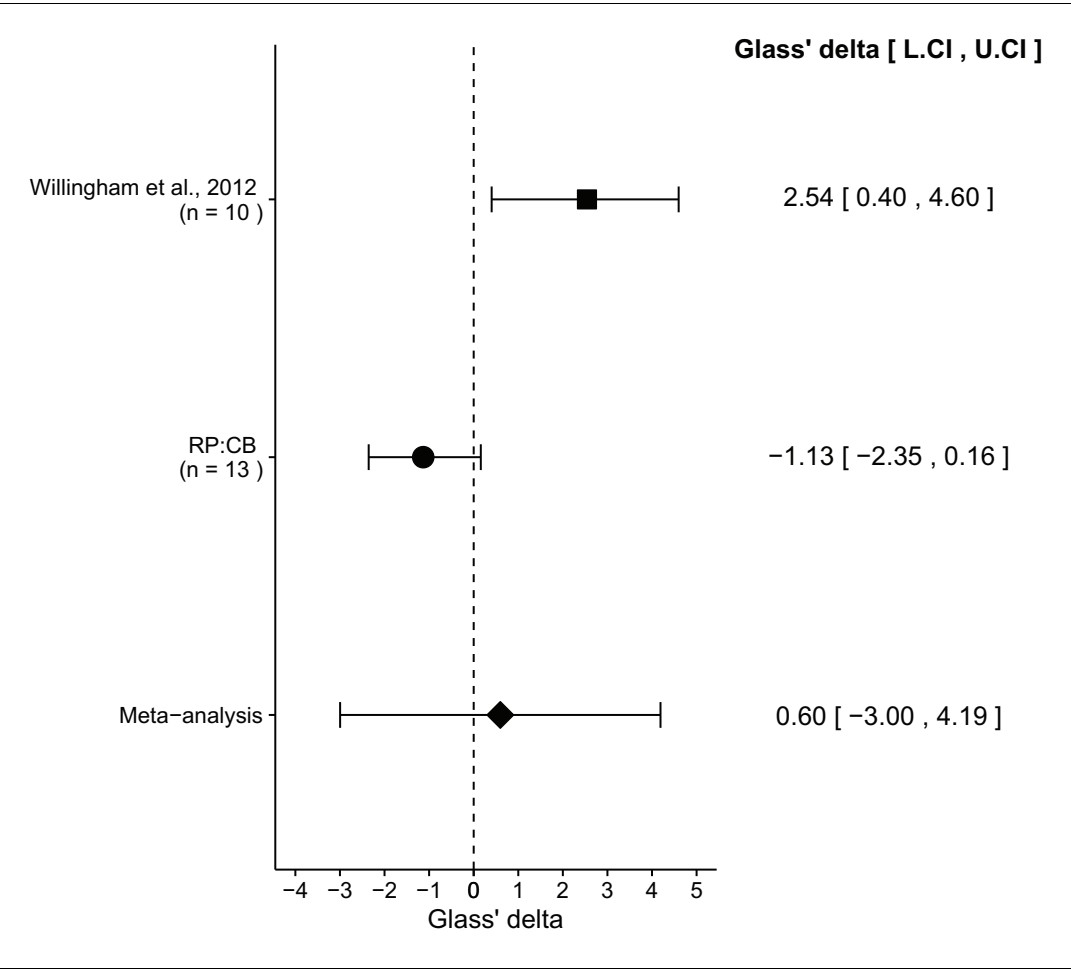

**Figure 3.** Meta-analysis of effect. Effect size (Glass' Δ) and 95% confidence interval are presented for *Willingham et al. (2012)*, this replication attempt (RP:CB), and a meta-analysis to combine the two effects of tumor weight comparisons. Sample sizes used in *Willingham et al. (2012)* and this replication attempt are reported under the study name. Random effects meta-analysis of tumors treated with IgG compared to anti-CD47 (meta-analysis $p$=0.745). Additional details for this meta-analysis can be found at https://osf.io/ha2bx/.

## Materials and methods

As described in the Registered Report (*Chroscinski et al., 2015*), we attempted a replication of the experiments reported in Figure 6A–C and Supplemental Table S4 of *Willingham et al. (2012)*. A detailed description of all protocols can be found in the Registered Report (*Chroscinski et al., 2015*). Additional detailed experimental notes, data, and analysis are available on the Open Science Framework (OSF) (RRID: SCR_003238) (https://osf.io/9pbos/; *Horrigan et al., 2016*).

### Cell culture

MT1A2 cells (shared by Weissman lab, Stanford University) were maintained in Dulbecco's Modified Eagle's Medium (DMEM) supplemented with 4 mM L-glutamine, 10% Fetal Bovine Serum (FBS) (Sigma-Aldrich, cat # F0392), 100 U/ml penicillin, and 100 µg/ml streptomycin. All cells were grown at 37°C in a humidified atmosphere at 5% $CO_2$.

Quality control data for the MT1A2 cell line are available on the OSF (https://osf.io/9r5hy/). This includes results confirming the cell line was free of mycoplasma contamination and common mouse pathogens. Additionally, STR DNA profiling of the cell line was performed.

## Therapeutic antibodies

Mouse anti-CD47, MIAP410 antibody (Weissman lab, Stanford University) and mouse IgG endotoxin depleted Protein A purified protein (Innovative Research, cat # IR-MS-GF-ED, lot # 111214DG, RRID: AB_1501657) were diluted with ice-cold PBS to a final concentration of 4.0 mg/ml and aliquoted into 18 vials of 0.8 ml each and stored at 4°C. One vial was used for each injection day and any remainder discarded. An enzyme-linked immunosorbent assay (ELISA) was performed with the mouse anti-CD47, MIAP410 antibody on plates coated with human CD47-Fc or mouse CD47-Fc (shared by Weissman lab, Stanford University). ELISA protocol details and data are available at https://osf.io/werk5/.

## Animals

All animal procedures were approved by the Noble Life Sciences IACUC# 15-05-001SCI and were in accordance with Noble Life Sciences policies on the care, welfare, and treatment of laboratory animals, which adhere to the regulations outlined in the USDA Animal Welfare Act (9 CFR Parts 1,2, and 3) and the conditions specified in the Guide for the Care and Use of Laboratory Animals (*National Research Council (US) Committee for the Update of the Guide for the Care and Use of Laboratory Animals, 2011*).

Mice were offered Certified Rodent Diet (Harlan Teklad, cat # 2018) *ad libitum*. The animal room was set to maintain between 19–22°C, a relative humidity of 40–65%, and a 12 hr light/dark cycle, which was interrupted for study-related activities.

A total of 20, six-eight week old female FVB mice (Charles River, strain code 207) were inoculated orthotopically with 50,000 MT1A2 cells at a density of $5 \times 10^4$ cells in 100 µl of FACS buffer with 25% vol/vol high concentration Matrigel (BD/Corning, cat # 354262, lot # 4090005) in the left abdominal mammary #4 fat pad. Mice were monitored every other day for signs of tumor growth. Once tumor growth was detected in any animal, tumors were measured using a digital caliper and body weights recorded. At the time of randomization, 17 animals had palpable tumors, with the 14 animals having the largest tumors entered into the study. Mice were ranked according to tumor volume from highest to lowest and the 14 animals with the largest tumors were assigned to a group using an alternating serpentine method. Designation of IgG or anti-CD47 to group 1 or 2 was done by randomly assigning the two treatment groups into one block using www.randomization.com (seed number = 21473) with variability of tumor volume measurements evenly distributed among the two conditions (Student's *t*-test; $t(12) = 0.44$, $p=0.67$). The remaining six mice not randomized to treatment were used to generate baseline readings for hematological analysis with one of these animals developing a tumor before the blood analysis was performed, which means of the 20 mice inoculated, 18 developed tumors. An initial attempt with 14 mice inoculated with MT1A2 cells was terminated because too few animals (10 out of 14 animals) had established tumors. Further details can be found in the 'Deviations from Registered Report' section and at https://osf.io/zch4n/.

Starting on the day of randomization and every other day for 30 days, 400 µg of anti-CD47 or IgG in a 100 µl volume was injected into the mammary fat pad approximately 2 mm proximal to tumor with a 30-gauge needle and 0.5 ml syringe. One animal receiving mouse anti-CD47 antibody was found dead, with no visible cause of death, before the end of the study. No body weight loss or behavioral changes were noted in any animals. At the end of the treatment period, on study day 31, animals were anesthetized with isoflurane, sacrificed, and tumors were excised, cleaned of surrounding fat tissue, and weighed. Tumor volume was calculated from caliper measurements using the formula (volume = 1/2(length*width$^2$) or calculated from weight of excised tumors and density (1.05 g/ml) (*Jensen et al., 2008*).

## Hematological analysis

Five days after the beginning of injections mice were anesthetized with isoflurane and 0.1–0.2 ml of blood was collected from the retro-orbital sinus using a microcapillary pipet into a collection tube containing kEDTA. Samples were rocked gently and analyzed within 2 hr using a Hemavet 950FS (Drew Scientific, Miami Lakes, Florida) using the built-in mouse program to determine complete blood count parameters (15 total parameters reported). Hematology profiles are available at https://osf.io/ucxwj/.

## Histopathology

Excised tumors were formalin-fixed, paraffin blocked, sectioned (two 5-micron thick sections of each tumor), and stained with hematoxylin and eosin as described in the Registered Report (*Chroscinski et al., 2015*). Sections were blindly examined microscopically by Alexander DePaoli, DVM, PhD (IDEXX Laboratories, Inc.) utilizing the severity score for inflammatory cell infiltrates (*Demaria et al., 2001*) where: absent = 0, minimal = 1, moderate = 2, brisk = 3. Histopathology report is available at https://osf.io/xky96/. H&E stained tumor sections are available at https://osf.io/43jau/. Remaining tumor sections are available upon request.

## Statistical analysis

Statistical analysis was performed with R software (RRID: SCR_001905), version 3.3.1 (*R Core Team, 2016*). All data, csv files, and analysis scripts are available on the OSF (https://osf.io/9pbos/). Confirmatory statistical analysis was pre-registered (https://osf.io/9gykv/) before the experimental work began as outlined in the Registered Report (*Chroscinski et al., 2015*). Data were checked to ensure assumptions of statistical tests were met. The hematological parameters were analyzed by multiple one-way ANOVAs; one for each of the 15 parameters analyzed. The Bonferroni correction, to account for multiple testings, was applied to the alpha error or the *p*-value. The Bonferroni corrected value was determined by divided the uncorrected value by the number of tests performed. For the alpha error this resulted in. 0033 (.05/15). A meta-analysis of a common original and replication effect size was performed with a random effects model and the *metafor* R package (*Viechtbauer, 2010*) (available at https://osf.io/ha2bx/). The original study data was shared by the original authors *a priori* during preparation of the experimental design. The data was published in the Registered Report (*Chroscinski et al., 2015*) and was used in the power calculations to determine the sample size for this study.

## Deviations from registered report

The type of high concentration Matrigel was different than what is listed in the Registered Report. The Registered Report listed the High Concentration Matrigel (BD/Corning, cat # 354248) while the replication experiment used the High Concentration, Phenol Red-Free Matrigel (BD/Corning, cat # 354262). The type of Matrigel used in the original experiment was not specified. The mouse IgG protein A purified protein used in this replication experiment was endotoxin depleted while the Registered Report did not indicate this purification methodology. This was clarified during communication with the original authors prior to performing the experiment. Additional materials and instrumentation not listed in the Registered Report, but needed during experimentation are also listed.

An initial attempt to inoculate 14 animals with MT1A2 cells as outlined in the Registered Report resulted in only 10 animals with established tumors. This was terminated because the predefined number of animals (7 per group) with established tumors was not reached. For the second attempt, which is reported here, the number of animals to inoculate with MT1A2 cells was increased to 20, based on the observed rate of engraftment in the first attempt. This attempt resulted in 17 animals with detectable tumors at the time of randomization, with the 14 animals having the largest tumors assigned to IgG or anti-CD47 treatment. The remaining 6 animals were used to generate baseline readings for hematological analysis with one of these animals developing a tumor before the blood analysis was performed, which means that 18 of the 20 mice inoculated developed tumors.

The Registered Report described 13 hematological parameters, similar to what was reported in *Willingham et al. (2012)* (Table S4), while this replication attempt reported 15 parameters. The difference stems from an inclusive measure of granulocytes used in the original study, while this replication attempt measured the number of the three principal types of granulocytes: neutrophils, basophils, and eosinophils.

## Acknowledgements

The Reproducibility Project: Cancer Biology would like to thank the Weissman lab for sharing critical reagents and data, specifically the MT1A2 cells, the anti-CD47, MIAP410 antibody, and the ELISA data. We thank Frank Graham and McMaster University for providing access to the MT1A2 cells. We would also like to thank the following companies for generously donating reagents to the

Reproducibility Project: Cancer Biology; American Type and Tissue Collection (ATCC), Applied Biological Materials, BioLegend, Charles River Laboratories, Corning Incorporated, DDC Medical, EMD Millipore, Harlan Laboratories, LI-COR Biosciences, Mirus Bio, Novus Biologicals, Sigma-Aldrich, and System Biosciences (SBI).

## Additional information

### Group author details

Reproducibility Project: Cancer Biology

Elizabeth Iorns: Science Exchange, Palo Alto, United States; Stephen R Williams: Center for Open Science, Charlottesville, United States; Nicole Perfito: Science Exchange, Palo Alto, United States; Timothy M Errington, http://orcid.org/0000-0002-4959-5143: Center for Open Science, Charlottesville, United States

### Competing interests

SKH: Noble Life Sciences Inc. is a Science Exchange associated lab. RP:CB: EI, NP: Employed by and hold shares in Science Exchange Inc. The other authors declare that no competing interests exist.

### Funding

| Funder | Author |
| --- | --- |
| Laura and John Arnold Foundation | Reproducibility Project: Cancer Biology |

The funder had no role in study design, data collection and interpretation, or the decision to submit the work for publication.

### Author contributions

SKH, Acquisition of data, Drafting or revising the article; RP:CB, Analysis and interpretation of data, Drafting or revising the article

### Ethics

Animal experimentation: All animal procedures were approved by the Noble Life Sciences IACUC# 15-05-001SCI and were in accordance with Noble Life Sciences policies on the care, welfare, and treatment of laboratory animals, which adhere to the regulations outlined in the USDA Animal Welfare Act (9 CFR Parts 1,2, and 3) and the conditions specified in the Guide for the Care and Use of Laboratory Animals (National Research Council (US) Committee for the Update of the Guide for the Care and Use of Laboratory Animals, 2011).

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
