## [Decision Letter]

Thank you for submitting your article "Replication Study: The CD47-signal regulatory protein alpha (SIRPa) interaction is a therapeutic target for human solid tumors" for consideration by *eLife*. Your article has been reviewed by four peer reviewers, and the evaluation has been overseen by a Reviewing Editor and Tadatsugu Taniguchi as the Senior Editor. The reviewers have opted to remain anonymous.

In your response before the decision you offered the view that the authors specifically do not indicate whether the results from this replication attempt should be used as evidence for or against the originally reported effect, but rather leave that to the scientific community to discuss and expound upon. However, we note that the whole purpose of the replication study is to provide conclusive evidence for or against the originally reported claims. With inconclusive evidence, the scientific community would have nothing to expound upon.

For publication in *eLife* the results must be conclusive. Editorially we are willing to consider one last round of revision of your manuscript. However, we want to be clear that *eLife* will consider a revised manuscript only if it contains enough independent experiments, replicates and internal controls, to arrive at conclusive results. You would need to test therapeutic effects with control tumors that are allowed to grow much larger (e.g. at least 1cm in diameter) than before. You would also need to take into account that responses to immunotherapies in mouse models (and in the clinic) can be delayed.

In sum, you need to provide sufficient experimental evidence to render the replication study conclusive. *eLife* would consider a revised manuscript only if and when this is achieved.

The original reviews are included below for your consideration.

Reviewer #1:

Errington et al.'s work represents an attempt to reproduce one experiment (out of a series of experiments) originally conducted by Willingham et al. The results of the reproducibility study presented herein not only differ from those of the original, but could also be interpreted to contradict the key findings of Willingham et al. Whether Errington et al.'s results themselves are reproducible remains to be seen, and I argue here that the findings from this group should be taken with a grain of salt.

Reproducibility aside, one necessary condition for conducting a preclinical study comprises the use of a robust model system. Plainly stated, a robust preclinical study wields an ability to yield a reasonable and interpretable result in the face of experimental variability that may be endogenously or exogenously introduced. The results proffered by Errington et al. lend strong credence to the notion that they are not operating with a robust experimental system, thereby leaving me to doubt their results. Specifically, there is tremendous variability in their tumor engraftment, as evinced by their first (and failed) attempt to engraft, followed by their second endeavor in which out of 20 mice, a mere 14 were usable for the study. Moreover, Errington et al. report inconsistent growth kinetics among the 14 mice that were engrafted with the same amount of cells. The fact that Errington et al. seemed to have so much trouble setting up the experiment leads me to question the results emanating from such a study. Simply put, Errington et al.'s study is neither robust due to the extreme variability in their starting points nor reproducible (as they themselves admit), and should not be taken as hardline contrarian evidence against the study by Willingham et al.

In summary, Errington et al. should repeat the study but should first show that they can achieve consistent engraftment of these tumor cells.

Reviewer #2:

Stephen Horrigan and Timothy Errington report their data in the reproducible project related to the paper by Willingham and others in 2012 entitled "The CD 47-signal regulatory protein alpha interaction is a therapeutic target for human solid tumors". The experiment reproduced was a murine clinical trial of an antibody to CD47 for the treatment of breast cancers in a syngeneic orthotopic model. Three parameters were evaluated: Tumor size, infiltration by immune cells into the tumors, and effects on blood counts.

Effects on blood counts, which are considered a toxicity, were reproduced, but the therapeutic effects were not, when compared to the original report. Indeed, in the current reproduction of the experiments, the control IgG was more therapeutically active than the treatment CD47 IgG, although not statistically significant.

There are some concerns in interpreting the therapeutic data.

1) The on target toxicity profile was similar to the original report, which is consistent with the mechanism of action of the CD47 IgG being present in the current model, yet the tumors did not shrink in these mice relative to control. This is an inconsistency. The tumors evaluated were very small. At the time of evaluation of the tumors, 30 days after the treatments began, approximately half of the tumors had weights between 10 and 60 mg, three of the tumors in the control group appeared to be only 10 mg or less in weight. This would correspond to tumors that were only about 2×2×3 mm. With tumors this small I wonder what accuracy of measurement and precision of surgery can be achieved? The methods state that 50,000 cells were injected orthotopically and that the treatment experiment would begin when the tumors were palpable. According to the protocol, the trial was terminated at 30 days. If the tumors were this small after 30 days, how big were the tumors at day 0, at the time of initial palpation? If they were on the order of the same size at that seen 30 days later, this suggests that the tumors grew little in the model, confounding the results; this would be true even in the case of the largest tumors that were observed at one month. This also seems in contrast to the rapid growth of the 50,000 cells injected to reach a "palpable tumor "over the 7-10 days expected, as stated in the Chroscinski registered report that describes the experiments to be done. (This is about a 100x increase in 7-10 days). In addition, there is considerable variability in tumor growth in both groups ranging from five-fold to 20-fold, respectively, between the smallest and the largest tumors. In order to better evaluate these data it would be important to know caliper sizes of all of the tumors at time 0 and any other tumor size measurements that were conducted in the 30 days before the animals were sacrificed. This would allow a better understanding of the growth kinetics and conclusions. Should these issues be discussed in the paper?

2) A second concern is that the authors report that there was a relatively low take rate for this tumor. In the "deviations" section it reports that an initial attempt to inoculate 14 animals with tumor was terminated because of the reduced establishment of tumors. In the "methods" section it states that an initial attempt with 17 mice inoculated with tumor was terminated because of too few animals with tumors. It is not clear why there is a discrepancy in these 2 sites of the paper; or did 2 attempts fail? It also raises the question that if tumor uptake rates were low, and tumor growth rates were slow (as discussed above,) and rates variable (as discussed above,) could this confound the results? These sorts of issues might be discussed in the paper.

Reviewer #3:

This is a well-written paper to replicate the CD47-SIRPa interaction reported by Willingham et al., 2012. The authors have done a nice job of describing their experiments. I recommend accepting this paper after the authors have addressed the comments given below.

Some of the panels in Figure 1 are difficult to read. As a good data reporting practice, I suggest the authors use the observed range of data for the y axis. For example, panel J of Figure 1 shows y-axis from 0 to 80, while the data seem to be between ~50 and ~60. Please use this range instead of 0 to 80. Similar comment applies to each panel in Figure 1.

The text needs to give more insights into Bonferroni correction i.e., how many tests are done and how the p-values were adjusted to get the Bonferroni-corrected p-values.

Given that this is a reproducibility project, the webpage https://osf.io/9pbos/ can provide a better documentation of the data and R codes used to generate all the figures, to obtain Bonferroni corrected p-values, and all the analyses reported. The current documentation is inadequate. For example, the file "Table S4 2 way ANOVA.R" shows R codes for 13 boxplots, but Figure 1 has 15 boxplots. At the top of this file there are objects called "y", "A" and "B", but there is no annotation explaining what is y and what are A and B. Please provide detailed documentation of how each Figure was obtained and for each analysis reported in the paper by providing well-annotated R functions.

The paper lacks a discussion on some thoughts from the authors about potential reasons why their results reported in Figure 2 show an opposite direction to that reported in Figure 6B of Willingham et al., 2012.

Reviewer #4:

This reproducibility project study sought to replicate experiments from Figure 6 A-C and Table S4 of Willingham et al., 2012, showing that treatment of immune competent mice bearing orthotopic M1A2 breast tumors with an anti-CD47 antibody clone MIAP410 led to tumor regression. While the authors of this replication study found that anti-CD47 antibody treatment induced anemia and moderate infiltration of tumors by neutrophils, it had no effect to inhibit tumor growth. Treated tumors on average were actually larger than IgG-treated controls, but there was no statistically significant difference between the two groups. The findings of this study are convincing and unfortunately do not support the major finding from the previous report that treatment with this anti-CD47 antibody induces tumor regression. The key reagents used in this study were well controlled – the cells, for example, were obtained from the author of the original study. As antibody was administered through injection into the fat pad, and no effect on tumor growth was found, it would be of interest to further demonstrate the presence of the antibody within treated tumors, for example by immunostaining tumor sections.

[Editors' note: further revisions were requested prior to acceptance, as described below.]

Thank you for resubmitting your work entitled "Replication Study: The CD47-signal regulatory protein alpha (SIRPa) interaction is a therapeutic target for human solid tumors" for further consideration at *eLife*. Your revised article has been evaluated by Tadatsugu Taniguchi (Senior Editor), a Reviewing Editor, and two reviewers.

We appreciate the changes you've made and we are moving ahead with acceptance, but we would recommend some further changes to the text first. In particular, we would like the Abstract and the Discussion to reflect the fact that, due to technical issues, the replication study itself is inconclusive. In the Abstract, we would suggest that the sentence that starts "The weight of tumors after 30 days […] " be revised to read as follows:

"However, our efforts to replicate the experiments in the original study which showed that tumor growth was inhibited by anti-CD47 treatment (Figure 6C; Willingham et al., 2012) were inconclusive.

The Discussion should also be revised to make this clear.

Also, the sentence that starts "Both IgG and anti-CD47 treated tumors resulted in[…]", should be revised to make the difference between the replication results and the original results clearer (e.g., please compare the results for lymphocytic infiltrate first, and then compare the results for neutrophilic infiltration).

Please revise your Abstract and Discussion accordingly. We have included the re-reviews below, but you do not need to respond to these comments when you resubmit.

Reviewer #2:

The authors have thoughtfully addressed my major concerns with changes to the data and interpretation, additional figures, additional statistical description. The discussion is more balanced and includes additional caveats to interpretation. With all of the caveats of conducting such a study, drawing definitive conclusions from one study alone (as noted by the authors) is difficult.

Reviewer #4:

While the discussion of the findings has been improved in the revised manuscript, the apparent issues with the tumor model persist as no new experimentation has been added. The previous concerns about the relative inability of the authors to achieve reproducible tumor growth using this model still casts doubt over whether or not the current studies offer solid evidence to contradict the published effects of antiCD47 antibody administration. It seems that for a reproducibility study to present data that is marginally robust, and then to leave the significance of the findings up to the community, this falls short of what should be required for refuting published work, as indicated in the last round of review.

---

## [Author Response]

*In your response before the decision you offered the view that the authors specifically do not indicate whether the results from this replication attempt should be used as evidence for or against the originally reported effect, but rather leave that to the scientific community to discuss and expound upon. However, we note that the whole purpose of the replication study is to provide conclusive evidence for or against the originally reported claims. With inconclusive evidence, the scientific community would have nothing to expound upon.*

Our previous comment was: “Finally, we specifically do not indicate whether the results from this replication attempt should be used as evidence for or against the originally reported effect, but rather leave that to the scientific community to discuss and expound upon.” We made this in reference to the replication outcome based on reviewers comment, but this is true of the original study as well. No single study can provide conclusive evidence for or against an effect, but rather it’s the cumulative evidence of multiple experiments and studies that provide the foundation of scientific claims. We outlined this in our article introducing the project. We have added a final paragraph in the Discussion to help clarify what this replication is capable of understanding as well as possible factors, as described in the reviewer comments below, that could impact the outcomes of this study.

*For publication in* eLife *the results must be conclusive. Editorially we are willing to consider one last round of revision of your manuscript. However, we want to be clear that eLife will consider a revised manuscript only if it contains enough independent experiments, replicates and internal controls, to arrive at conclusive results. You would need to test therapeutic effects with control tumors that are allowed to grow much larger (e.g. at least 1cm in diameter) than before. You would also need to take into account that responses to immunotherapies in mouse models (and in the clinic) can be delayed.*

*In sum, you need to provide sufficient experimental evidence to render the replication study conclusive. eLife would consider a revised manuscript only if and when this is achieved.*

This replication attempt, like all of the replication attempts in this project, are designed to perform independent replications with a calculated sample size to detect the originally reported effect size with at least 80% power. Further, this project will report the cumulative evidence across multiple independent replications among multiple studies. Thus, no single replication from this project, just like no original experiment or study, can provide conclusive evidence for or against an effect; rather, it’s the cumulative evidence that forms the foundation of scientific knowledge. However, we understand the desire to perform the experiment independently again, but with modifications to the design outlined and peer reviewed in the Registered Report – before the results were known. While, it’s not within the scope of this project, or as part of this publishing model, to also conduct these studies, the results of this replication bring variables not previously thought to influence the experiment into question (size of the control tumors at the end of the study, length of treatment, etc). Importantly though, it is only because of the results that these and other aspects now become targets for hypothesizing and investigation.

*The original reviews are included below for your consideration.*

Reviewer #1:

*Errington et al.'s work represents an attempt to reproduce one experiment (out of a series of experiments) originally conducted by Willingham et al. [...] In summary, Errington et al. should repeat the study but should first show that they can achieve consistent engraftment of these tumor cells.*

We agree the engraftment rate reported in the manuscript was not 100%, however it is unknown to us what the rate was in the original study. The initial attempt was terminated because the prespecified number of animals with detectable tumors to enroll in the study (7 per group) was not achieved, with only 10 out of 14 mice developing tumors. In the second attempt, which inoculated 20 animals based on the observed rate of engraftment in the first attempt, 17 had detectable tumors at the time of randomization, with only 14 mice (with the largest tumor sizes) being enrolled in the study as prespecified in the Registered Report. The remaining 6 mice were used for the baseline hematological measurement, with another one of these developing a tumor before the blood analysis was performed, which means of 20 mice inoculated 18 developed tumors. Variability of tumor volume measurements at the time of assignment were evenly distributed among the two conditions during the randomization. We have revised the manuscript to include the test of tumor volume distribution among the two cohorts.

We agree there is a difference in the range of tumor weights reported in this study, compared to the original work, but there are many factors that could account for discrepancies between the studies. These include factors such as genetic drift and heterogeneity in the cell line population, microbiome of recipient mice, circadian biological responses to therapy, etc., which we can include in the discussion of a revised manuscript. We also agree performing another attempt of this experiment would begin to explore if these, or other, factors influence the outcome of this study. However, what we reported in this manuscript, following the protocol reviewed before these results were known, would be valuable to that effort, whether we or others conducted another attempt. Finally, we specifically do not indicate whether the results from this replication attempt should be used as evidence for or against the originally reported effect, but rather leave that to the scientific community to discuss and expound upon.

In the revised manuscript we have further clarified the engraftment numbers observed in this replication attempt. We have also added a final paragraph in the Discussion to describe possible factors that could impact the outcomes of this study.

Reviewer #2:

*Stephen Horrigan and Timothy Errington report their data in the reproducible project related to the paper by Willingham and others in 2012 entitled "The CD 47-signal regulatory protein alpha interaction is a therapeutic target for human solid tumors". The experiment reproduced was a murine clinical trial of an antibody to CD47 for the treatment of breast cancers in a syngeneic orthotopic model. Three parameters were evaluated: Tumor size, infiltration by immune cells into the tumors, and effects on blood counts.*

*Effects on blood counts, which are considered a toxicity, were reproduced, but the therapeutic effects were not, when compared to the original report. Indeed, in the current reproduction of the experiments, the control IgG was more therapeutically active than the treatment CD47 IgG, although not statistically significant.*

*There are some concerns in interpreting the therapeutic data.*

*1) The on target toxicity profile was similar to the original report, which is consistent with the mechanism of action of the CD47 IgG being present in the current model, yet the tumors did not shrink in these mice relative to control. This is an inconsistency. The tumors evaluated were very small. At the time of evaluation of the tumors, 30 days after the treatments began, approximately half of the tumors had weights between 10 and 60 mg, three of the tumors in the control group appeared to be only 10 mg or less in weight. This would correspond to tumors that were only about 2×2×3 mm. With tumors this small I wonder what accuracy of measurement and precision of surgery can be achieved? The methods state that 50,000 cells were injected orthotopically and that the treatment experiment would begin when the tumors were palpable. According to the protocol, the trial was terminated at 30 days. If the tumors were this small after 30 days, how big were the tumors at day 0, at the time of initial palpation? If they were on the order of the same size at that seen 30 days later, this suggests that the tumors grew little in the model, confounding the results; this would be true even in the case of the largest tumors that were observed at one month. This also seems in contrast to the rapid growth of the 50,000 cells injected to reach a "palpable tumor "over the 7-10 days expected, as stated in the Chroscinski registered report that describes the experiments to be done. (This is about a 100x increase in 7-10 days). In addition, there is considerable variability in tumor growth in both groups ranging from five-fold to 20-fold, respectively, between the smallest and the largest tumors. In order to better evaluate these data it would be important to know caliper sizes of all of the tumors at time 0 and any other tumor size measurements that were conducted in the 30 days before the animals were sacrificed. This would allow a better understanding of the growth kinetics and conclusions. Should these issues be discussed in the paper?*

We agree that presenting the tumor volumes determined from the caliper measurements would be valuable in the manuscript. We generated an additional figure (Figure 2—figure supplement 1) to present this data. We have revised the manuscript to comment on the spread of the reported values observed in this replication attempt in context to the spread of values reported in Willingham et al., 2012. Further, we have revised the manuscript to expand upon other published reports using this model. While we did not find other studies, besides the original study and this replication, performing the assay the same way (orthotopic injections of 50,000 cells), we commented on other in vivo experiments utilizing MT1A2 cells and the approximate growth kinetics (Ahn and Brown, 2008; Noblitt et al., 2005). Interestingly, one paper (Desilva et al., 2012) briefly stated why they purposefully did not utilize this model in their study:

“Furthermore, in our hands spontaneous regression of MT1A2 tumors occurred at a sufficiently high frequency to confound efficacy studies."

The variability in tumor growth could be due to many factors (such as ones described in response to Reviewer #1). The variability should also be viewed in the context that cell growth is exponential under an ideal scenario. There are factors that influence and alter the growth of the tumor initially compared to the continued growth of the tumor in vivo, such as availability of nutrients, oxygen, and space. These points have been included in the Discussion of the revised manuscript.

*2) A second concern is that the authors report that there was a relatively low take rate for this tumor. In the "deviations" section it reports that an initial attempt to inoculate 14 animals with tumor was terminated because of the reduced establishment of tumors. In the "methods" section it states that an initial attempt with 17 mice inoculated with tumor was terminated because of too few animals with tumors. It is not clear why there is a discrepancy in these 2 sites of the paper; or did 2 attempts fail? It also raises the question that if tumor uptake rates were low, and tumor growth rates were slow (as discussed above,) and rates variable (as discussed above,) could this confound the results? These sorts of issues might be discussed in the paper.*

Thank you for raising this point. There is an error in the methods section when describing the initial attempt to establish tumors in the mice. There was only one initial attempt with 14 mice (as specified in the Registered Report), not 17. This has been corrected in the revised manuscript.

The initial attempt was terminated because the prespecified number of animals with detectable tumors to enroll in the study (7 per group) was not achieved, with only 10 out of 14 mice having tumors. In the second attempt, which increased the number of mice inoculated to 20 based on the observed rate of engraftment in the first attempt, 17 had detectable tumors at the time of randomization. Only 14 mice, with the largest tumor sizes, were enrolled in the study, with the remaining mice used for the baseline hematological measurement. As described in the response to Reviewer #1, we are unaware of what the tumor take rate was in the original study. In the revised manuscript we have further clarified the engraftment numbers observed in this replication attempt. We have also added a final paragraph in the Discussion to describe possible factors that could impact the outcomes of this study.

Reviewer #3:

*This is a well-written paper to replicate the CD47-SIRPa interaction reported by Willingham et al., 2012. The authors have done a nice job of describing their experiments. I recommend accepting this paper after the authors have addressed the comments given below.*

*Some of the panels in Figure 1 are difficult to read. As a good data reporting practice, I suggest the authors use the observed range of data for the y axis. For example, panel J of Figure 1 shows y-axis from 0 to 80, while the data seem to be between ~50 and ~60. Please use this range instead of 0 to 80. Similar comment applies to each panel in Figure 1.*

We have revised Figure 1 to only show the range the observed data fall within as suggested. Part of the reason a wider range was included was to show how the values relate to the normal range (grey region of graph). To also present this aspect of the data, we have revised this data into a table format showing the normal expected range and made it a figure supplement (Figure 1—figure supplement 1).

*The text needs to give more insights into Bonferroni correction i.e., how many tests are done and how the p-values were adjusted to get the Bonferroni-corrected p-values.*

We agree and have included this in ‘Statistical analysis’ section in the methods of the revised manuscript.

*Given that this is a reproducibility project, the webpage https://osf.io/9pbos/ can provide a better documentation of the data and R codes used to generate all the figures, to obtain Bonferroni corrected p-values, and all the analyses reported. The current documentation is inadequate. For example, the file "Table S4 2 way ANOVA.R" shows R codes for 13 boxplots, but Figure 1 has 15 boxplots. At the top of this file there are objects called "y", "A" and "B", but there is no annotation explaining what is y and what are A and B. Please provide detailed documentation of how each Figure was obtained and for each analysis reported in the paper by providing well-annotated R functions.*

We have increased the annotation and documentation of the files presented online. In regard to the example file, “Table S4 2-way ANOVA.R”, it was used in the power calculations associated with the published Registered Report and contains the original data, not the replication data. We have edited these older files to clearly distinguish the difference. As such, the difference in boxplots is because the original study had 13 parameters, while this replication had 15 parameters due to differences in instrumentation.

*The paper lacks a discussion on some thoughts from the authors about potential reasons why their results reported in Figure 2 show an opposite direction to that reported in Figure 6B of Willingham et al., 2012.*

We agree and having included a discussion on some of the known variables that were not specifically addressed or could not be easily controlled in this replication attempt.

Reviewer #4:

*This reproducibility project study sought to replicate experiments from Figure 6 A-C and Table S4 of Willingham et al., 2012, showing that treatment of immune competent mice bearing orthotopic M1A2 breast tumors with an anti-CD47 antibody clone MIAP410 led to tumor regression. While the authors of this replication study found that anti-CD47 antibody treatment induced anemia and moderate infiltration of tumors by neutrophils, it had no effect to inhibit tumor growth. Treated tumors on average were actually larger than IgG-treated controls, but there was no statistically significant difference between the two groups. The findings of this study are convincing and unfortunately do not support the major finding from the previous report that treatment with this anti-CD47 antibody induces tumor regression. The key reagents used in this study were well controlled – the cells, for example, were obtained from the author of the original study. As antibody was administered through injection into the fat pad, and no effect on tumor growth was found, it would be of interest to further demonstrate the presence of the antibody within treated tumors, for example by immunostaining tumor sections.*

We agree it would be of interested to further explore if the antibody is detectable within the tumor sections. While we feel this is beyond the scope of this Replication Study, we think making it possible for others interested in examining this or other possible exploratory avenues is worthwhile. We suggest making available the material that can be shared by the lab, in this case the remaining tumor sections that were formalin-fixed, paraffin blocked, and sectioned. We have revised the manuscript to indicate what material is available.

[Editors' note: further revisions were requested prior to acceptance, as described below.]

*We appreciate the changes you've made and we are moving ahead with acceptance, but we would recommend some further changes to the text first. In particular, we would like the Abstract and the Discussion to reflect the fact that, due to technical issues, the replication study itself is inconclusive. In the Abstract, we would suggest that the sentence that starts "The weight of tumors after 30 days […]" be revised to read as follows:*

*"However, our efforts to replicate the experiments in the original study which showed that tumor growth was inhibited by anti-CD47 treatment (Figure 6C; Willingham et al., 2012) were inconclusive.*

*The Discussion should also be revised to make this clear.*

*Also, the sentence that starts "Both IgG and anti-CD47 treated tumors resulted in […]", should be revised to make the difference between the replication results and the original results clearer (e.g., please compare the results for lymphocytic infiltrate first, and then compare the results for neutrophilic infiltration).*

*Please revise your Abstract and Discussion accordingly.*

We have revised the Abstract and Discussion sections to explicitly state the replication study is confounded by the spontaneous regression observed. The Abstract has also been revised to clearly distinguish the replication and original results for lymphocytic infiltrate and neutrophilic infiltration.